# Protecting Wastewater Workers by Categorizing Risks of Pathogen Exposures by Splash and Fecal-Oral Transmission during Routine Tasks

Rasha Maal-Bared

Quality Assurance and Environment, EPCOR Water Services Inc., EPCOR Tower, 2000, 10423-101 Street NW, Edmonton, AB T5H 0E8, Canada; rmaalbar@epcor.com; Tel.: +1-780-412-7617

**Abstract:** Quantitative microbial risk assessments (QMRAs) present an opportunity to systematically assess risk to wastewater treatment plant (WWTP) workers and mitigate work-related infectious diseases. However, while QMRAs often explore the impacts of aeration or treatment mechanism, or the use of controls to mitigate risk (e.g., ventilation, personal protective equipment (PPE)), fewer studies address other variables, such as differing tasks across plants, time spent conducting these tasks or size of plant. QMRA approaches also vary substantially in their findings and recommendations. The objective of this paper is to provide a risk-based wastewater worker task characterization for urban, municipal and industrial WWTPs along with mitigation measures. Routine tasks fell into five categories in ascending order of exposure and risk, Type A being the lowest and Type E being the highest. Percentage of full-time equivalent time spent on each task category was estimated, along with amount of wastewater exposure (mL) and inhalation duration (h). Estimates differed between urban and municipal plants but were similar in industrial and municipal systems. Finally, a checklist was developed to identify potential mitigation measures and prioritize H&S solutions for eight inspected WWTPs. The present work provides practical information for job safety assessments, H&S policies and QMRA method refinement.

**Keywords:** biohazards; risk; exposure assessment; QMRA; pathogens; health and safety; occupational health protection; wastewater workers

## 1. Introduction

Wastewater treatment and collection systems provide an essential service for the protection of public health and the environment. Domestic wastewater is a combination of human feces, urine and graywater that is used in households for washing, bathing, cleaning and meal preparation [1]. Since the average human excretes about 100–500 g of feces and about 1–1.3 L of urine per day [2], it is inevitable that domestic wastewater will contain a broad range of bacteria, viruses, protozoa and helminths that are shed by many symptomatic and asymptomatic carriers in the served population [3]. Thus, wastewater system workers who come into contact with wastewater or sludge during collection, treatment, laboratory analyses, sludge disposal, or plant maintenance and repair activities may be exposed to these biological agents. Such exposures could result in occupational infection and disease, if not properly controlled [4–9].

Research investigating wastewater worker occupational health dates back to the early 1900s [10,11] but the term sewage workers' syndrome was only coined in the 1970s [12,13]. While there has been sustained interest in assessing the impacts of exposures to chemical and biological hazards at wastewater treatment plants (WWTP), it was not until the COVID-19 pandemic that the number of papers using quantitative microbial risk assessment (QMRA) at WWTPs to assess risk of occupational infections in wastewater workers increased substantially [14–21]. These QMRAs explored the impacts of aeration or treatment mechanism (e.g., umbrella aeration tank versus microaeration, membranes) [14,16]

or the use of controls to mitigate risk, such as ventilation [16] and personal protective equipment (PPE) [16,17,20]. Fewer studies adequately addressed other variables, such as differing tasks across plants and time spent conducting these tasks (Cowie et al., 2008). Yan et al. (2021) divide exposure types into temporary entrants (researchers, visitors, and inspectors) and staffs (field engineer and laboratory technician) [17]. Similarly, Gui et al. (2022) divide exposure into academic visitors, field engineers and office staff and assign set exposure time and frequency regardless of assigned tasks [20]. Zhang et al. (2022) provide the most detailed description of tasks and exposures but do not provide details on corresponding WWTP size or mitigation strategies. Overall, these studies lack practical guidance for utility managers, industrial hygienists and Health and Safety (H&S) staff attempting to identify potential safety improvements [20].

QMRA presents an opportunity to systematically assess risk to wastewater treatment plant (WWTP) workers and mitigate work-related infectious diseases using a structured and systematic approach [22,23]. Thus, improving these assessments by providing a more detailed and nuanced exposure characterization of workers at WWTPs would be beneficial. The objective of this paper is to provide a wastewater worker task characterization for urban (>300 megaliters per day (MLD)), municipal (10 MLD) and industrial (0.001 MLD) WWTPs along with appropriate measure for mitigating risk against splash and fecal-oral transmission exposure.

## 2. Materials and Methods

### 2.1. Participating Wastewater Treatment Plants

The urban WWTP is a tertiary plant that serves approximately 1.1 million residents in Edmonton (Alberta, AB, Canada). The plant treats 310 million $m^3$ of wastewater per day with peak flows up to 910 million $m^3$ per day. The plant has about 140 employees in areas divided into: utility crew who perform clean up and upkeep tasks; maintenance crews who perform proactive and reactive maintenance; operations teams who conduct assessments, field testing, operational adjustments, visual checks, etc.; engineers; laboratory staff; subject matter experts (e.g., H&S, scientists) and visitors.

The municipal WWTP was much smaller and representative of many smaller systems in North America and Europe. It is a secondary plant with UV disinfection that serves a population of approximately 20,000 people. It treats 10 million $m^3$ per day. The plant employs 20 staff, who are more versatile and cover several areas due to limitation of resources (e.g., operations staff also conduct lab testing, engineers also act as operators).

The industrial WWTP is a tertiary system in an industrial area. It receives approximately 500 $m^3$ per day of influent from the surrounding industrial camps serving 200–500 workers. The secondary effluent is discharged to a polishing plant and the waste sludge moved for additional treatment at a secondary location. The plant has approximately 15 employees who run both the drinking water and wastewater operations at three different locations and work in weekly shifts.

### 2.2. Tasks and Activities Performed by WWTP Staff

In 2015, utility managers, team leads, WWTP workers and H&S employees from all representative exposure groups at urban, municipal and industrial plants were invited to participate in focus groups to aid with the categorization of tasks performed on a daily basis. Nine focus groups with equal representation across all areas (where possible) met over a three-month period. Each focus group had 8–15 participants (n = 119), who used the institutional standard operating procedures and hazard registries to develop a list of tasks performed on a routine basis. The same groups were asked to estimate the amount of time spent per task and the percentage of time spent in contact with untreated, partially treated or fully treated wastewater. The information collected with these focus groups was then used to define levels of exposure and provide a list of corresponding plant activities. The exposure data was represented in full-time personnel equivalents or full-time equivalent

(FTE), which measure how many total full-time employees and part-time employees add up to a full eight-hour daily work shift over a one-year period.

*2.3. Risk Mitigation and Management*

Focus group results were reviewed with H&S experts, workers and utility managers. A summary of potential control measures and industry standards were compiled through literature review, site visits and consultation with the experts. The literature review mainly relied on national and international wastewater worker protection standards (e.g., the US Center for Disease Control and Prevention, the Occupational Safety and Health Association, Work Safe Alberta, etc.), as the scientific literature did not focus on practical operational occupational disease prevention. In addition, the list of controls was matched to the identified exposure categories defined by the focus groups. The H&S experts created a WWTP occupational H&S microbial exposure mitigation checklist that was used to conduct eight site visits across different WWTPs in Alberta (municipal, urban, industrial). The site visits were used to document current hygiene and safety practices and evaluate how these sites could improve their current standards. The results were documented.

## 3. Results

*3.1. Tasks Resulting in Exposure to Wastewater*

The risk of contact with wastewater is highly task-dependent. Thus, routine tasks fell into several natural categories in ascending order of exposure and risk. The team associated each category with the number of full-time-equivalents (FTE) approximately committed for that activity. Five types of tasks were identified and described in Table 1.

**Table 1.** Categorization of types of activities performed by wastewater treatment workers and corresponding wastewater exposure and protection in ascending order of exposure risk.

| Activity Type | Description |
|---|---|
| Type A | Type A activities are primarily located within office spaces or on-site trailers. Typically, no controls are required for these activities beyond basic hygiene. No primary or secondary contact with wastewater is expected at this level |
| Type B | Type B includes activities that require walking through plant areas and may include inspections. Typical controls for these activities will include basic hygiene, safety glasses, hard hat, and safety boots. No primary contact with wastewater is expected at this level, though some secondary contact is possible through fomites, splashing or bioaerosols. |
| Type C | Type C activities involve primary—if minor—contact with some wastewater in the form of adjusting valves, working on wiring, isolating equipment, job setup, and other similar work. Typical controls for these activities will include hard hat, safety glasses, steel-toed boots, coveralls, gloves, hand washing, and other standard administrative controls (e.g., safe work plan, hazard assessments). For some tasks additional personal protective equipment (PPE), including a face respirator and portable gas monitor, may be deemed necessary. |
| Type D | Type D includes activities undertaken for lab analysis, sampling, pump maintenance, and disconnecting equipment. Moderate levels of primary contact with wastewater can be expected at this level. Typical controls for these activities include hard hat, safety glasses, steel-toed boots, gloves, hand washing, and other standard administrative controls. A portable gas monitor may at times also be required, depending on the task. |

**Table 1.** *Cont.*

| Activity Type | Description |
|---|---|
| Type E<br><br><br><br><br>Type E1 | Type E activities include working inside various tanks, lift stations, using hoses, doing emergency repairs, adjusting pressurized equipment or valves, and any other work with a high likelihood of splashing. These activities involve the greatest level of primary contact with wastewater. A wide range of controls may be activated for this activity level, depending on the specific tasks, including hard hat, safety glasses, steel-toed rubber boots, face mask, rubber rain suit or coveralls, gloves, and ear protection. A sub-type, Type E1, was created to cover the specific activity which could involve accidentally being splashed in the face with a larger quantity of wastewater. |

*3.2. Estimates of Worker Exposure to Liquid Wastewater and Aerosols by Task*

Table 2 provides estimated exposures per FTE per day for a given type of activity. Type A and B activities are contact with trace amounts of wastewater on railings and other surfaces (i.e., fomite transmission through shared surfaces given that lunchrooms, computers and work areas are shared); the volumes are very low but the contacts per day are quite high. Contact with liquid wastewater begins with Type C. In general, Type E activities see the highest exposures; Type E1 is a subset of Type E and involves a direct splash to the face, which is much rarer. In the FTE column for all WWTP types, we provide the percentage of the FTE shift allocated to this task category.

**Table 2.** Exposure Information including estimates for wastewater contact volumes and aerosol exposures for urban, municipal and industrial WWTP full time equivalents (FTE).

| Exposure Category | Urban | | | Municipal | | | Industrial | | |
|---|---|---|---|---|---|---|---|---|---|
| | FTE | Liquid Contact (mL) | Aerosol Contact (h) | FTE | Liquid Contact (mL) | Aerosol Contact (h) | FTE | Liquid Contact (mL) | Aerosol Contact (h) |
| Type A | 15% | 3.00 | - | 14% | 3.00 | - | 10% | 3.00 | - |
| Type B | 10% | 5.00 | 0.40 | 2% | 5.00 | 0.40 | 3% | 5.00 | 0.40 |
| Type C | 20% | 0.001 | 0.80 | 6% | 0.003 | 1.60 | 7% | 0.004 | 1.60 |
| Type D | 7% | 0.02 | 0.80 | 2% | 0.02 | 0.80 | 3% | 0.03 | 0.80 |
| Type E | 1% | 0.09 | 4.00 | <1% | 0.06 | 4.00 | <1% | 0.09 | 4.00 |
| Type E1 | 1% | 0.01 | - | <1% | 0.02 | - | <1% | 0.02 | - |

*3.3. Exposure Estimates by Activity and Plant Type*

Figure 1 provides a summary of the estimated exposures to wastewater as a function of plant type and type of activity. The exposures are expressed as the amount of time spent in each activity in FTE. The urban WWTP has the highest number of FTEs that work with wastewater, with the most time spent in Type A and C activities. For the municipal and industrial WWTP, Type A activities occupy a higher percentage of time because the staff are likely to work in both the drinking water and wastewater plants splitting their time. While Type E activities may be the highest risk activities, they account for the smallest fraction of total FTEs.

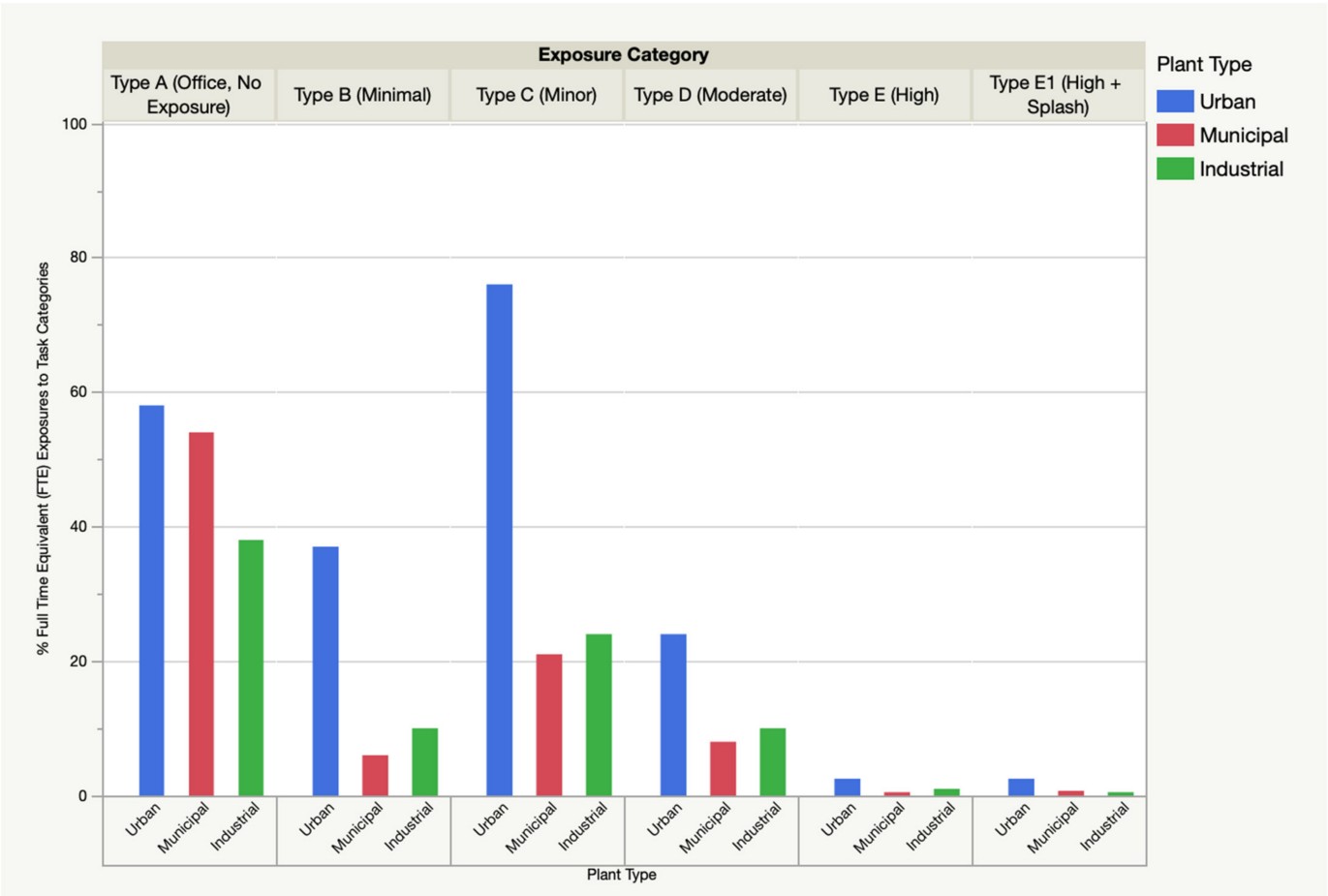

**Figure 1.** Exposure estimates for wastewater workers for urban, municipal and industrial WWTP full time equivalents (FTE).

*3.4. Comparing Current Industry Pratices to Recommended Control Measures*

The summary of potential control measures compiled through literature review, site visits and consultation with health and safety experts is presented in Table 3. The list followed the OSHA hierarchy of controls [24]. These controls can help when developing task-specific work plans while keeping in mind challenges that come with the operation and maintenance of small plants. We used the control measures from Table 3 to identify and outline the most suitable and practical controls for mitigation of risk by exposure category previously defined by the focus groups in Table 4. Some of the most common engineering controls deficiencies were the use of single lockers to store street and work clothes, the lack of showers for post-shift washup and the lack of designated eating and smoking areas. Most common administrative findings were inconsistent descriptions and applications of controls, and generic rather than task-specific controls. There were also variations in training programs and lack of obvious signage (e.g., PPE, designated eating or smoking areas). The hygiene findings included inconsistent hand washing practices and facilities, lack of designated cleaning areas, employees not changing out of their PPE before entering eating areas and meeting rooms, using single lockers for street and work clothes and placing dirty boots next to clean shoes throughout the shift. Finally, the most common PPE findings were related to the lack of a practical range of gloves that allow workers the dexterity needed to perform their jobs (e.g., leather, insulated rubber, nitrile), the use of face shields, dust masks and body protection for Type E tasks was unstandardized, type of footwear used varied and the use of respiratory protection was rarely identified as needed in the Job Safety Assessment (JSA).

**Table 3.** Current recommended industry practices and implemented control measures to minimize exposures to biohazards based on 8 WWTP evaluations using constructed checklists.

| Checklist Controls | Observations Based WWTP Visits | Potential Improvements |
|---|---|---|
| Engineering Controls | | |
| • Facility design<br>• Structural hygiene (lockers, sinks, showers, lunchrooms)<br>• Ventilation systems and air exchanges | Available hygiene facilities vary from plant to plant (sometimes operated and owned by different parties):<br>• Single lockers<br>• No showers<br>• Clean eating areas | • Frequent hand washing facilities that include nail brushes and disposable paper towels.<br>• Designated eating areas.<br>• Separate storage for work clothes and street clothes<br>• Access to showers<br>• Engineered solutions are suggested to avoid aerosol transmission- consider encasement, splash guard deflectors, wind baffles (redirect aerosol hazards).<br>• In order to reduce exposure to airborne microorganisms and endotoxins, conduct heavy equipment waste disposal operations in cab-sealed positive pressure air-conditioned cabs to filter air recirculating units. |
| Administrative Controls | | |
| • Procedures for tasks involving contact with wastewater<br>• Signage clean areas<br>• Hand washing posters<br>• Orientation<br>• Training | • Inconsistent descriptions of controls, and generic rather than task specific controls<br>• Some sites have found methods to eliminate exposure to splashes during cleaning (e.g., minimize use of hoses when inside basins)<br>• The training of hygiene practices vary from site to site<br>• Signage is variable | • System for reporting biohazard exposures (e.g., splashes)<br>• Ensure diphtheria/tetanus and Hep A immunizations are up-to-date and offer in the workplace<br>• Training on a regular basis on safe handling and disposal of biological hazards and safe use and maintenance of PPE.<br>• Policies and procedures applicable to occupational health and safety (OHS) practices reviewed and updated annually<br>• Cleaning soiled devices and PPE immediately after use<br>• Education signs regarding mandatory precautions while at work<br>• Literature available in languages other than English<br>• H&S pocket card indicating precautions employees need to take to reduce risk of infection and in case of accidental exposure |
| HYGIENE CONTROLS | | |
| • Hand washing<br>• Use of showers<br>• Use of separate lockers for work and street clothes<br>• Washer/Dryer or cleaning service for coveralls<br>• Clean area zones | • Hand washing practices and facilities inconsistent<br>• Designation/awareness of clean areas varies<br>• Employees not changing their work clothes before entering the eating area.<br>• Changing from work clothes to street clothes procedures not consistently being followed.<br>• Work clothes and street clothes placed in the same locker.<br>• Work shoes and daily work shoes adjacent | • Regular training program related to personal hygiene measures<br>• Sterile wipes for cleansing wounds and a supply of sterile, waterproof adhesive dressings<br>• Signage on mandatory precautions |

**Table 3.** *Cont.*

| Checklist Controls | Observations Based WWTP Visits | Potential Improvements |
|---|---|---|
| PPE Controls | | |
| • Safety glasses<br>• Steel toed rubber boots<br>• Face shield<br>• Dust mask<br>• Rubber rain suit/coveralls/Tyvek<br>• Gloves | • The current PPE being used at sites is inconsistent based on the hazards of the task.<br>• Gloves range from leather to cloth mechanics to nitrile.<br>• Use of Face Shields and Dust masks vary in Type E tasks<br>• Type of footwear varies<br>• Range of body protection used in Type E activities–full rain suits, partial rain gear, Tyvek suits<br>• Protection from aerosols rarely used even when job safety assessment conducted (JSA) | • In wet areas use impermeable or nitrile gloves, wrist to elbow coverage.<br>• In wet areas use Rubber rain suit/Tyvek coveralls<br>• Proper disposal of contaminated PPE.<br>• Use protection from aerosols–conduct respirator fit testing and use respirator code of practice when working around aerosols<br>• PPE to be determined by hazard identification |

**Table 4.** Recommended control measures by WWTP worker activity or task with examples.

| Activity/Task | Examples | Control Measure Guidelines |
|---|---|---|
| High contact with WW–splashing | Washing tanks, washing lift stations, using hoses to wash down equipment | Water resistant gear, face protection, possible respiratory protection, additional hygiene practices such as washing down or removing gear before moving into other areas (plant and field work). |
| Moderate contact with WW | Lab analysis, sampling, pump maintenance, disconnecting pipes, cleaning up small spills | Water resistant gloves, coveralls/lab coats (fabric), basic hygiene practices [1], potentially water resistant clothing/boots and additional hygiene practices depending on activity |
| Minor contact with WW | Adjusting valves, work on wiring, isolating equipment, job setup | Standard PPE [2], gloves (preferably with some water resistancy for certain activities), coveralls (fabric), basic hygiene practices [1] |
| Minimal contact | Inspections, walking around plants and worksites | Standard PPE [2], gloves, basic hygiene practices [1] |
| General Hygiene Facilities | | Clean eating areas, wash facilities available to clean after exposure to wastewater, separation of work and street clothes |
| Hygiene Practices | | Washing hands, signage, orientations |
| Infectious Disease Outbreaks | Suspected presence of a Class A pathogen (such as Ebola) | Stop any non-essential work; for activities with direct wastewater contact, use water resistant clothing, gloves, boots, face protection (face shied and dust mask), washing down & removing gear before moving into other areas |

[1] Basic hygiene usually includes hand washing after returning from plant areas and respecting clean areas such as lunchrooms. [2] Standard PPE includes safety glasses, steel toed boots and hard hats.

## 4. Discussion

In recent years, the number of papers using QMRA to assess the risk of occupational infections at WWTPs increased substantially [14–21]. This QMRA work has highlighted the need for a better understanding of contact exposure levels and pathways to support accurate cumulative risk assessments for wastewater and collection system workers [25]. Wastewater professionals working in the water sector are uniquely positioned to provide insight into these areas and contribute knowledge that QMRA professionals may not have access to. The objective of this paper was to characterize the types of routine activities at WWTPs, define levels of exposure and identify relevant controls needed to mitigate risk of infection of wastewater workers at urban, municipal and industrial WWTPs.

The main mode of transmission assessed was the fecal-oral route and mucous membrane contact although we did estimate amounts of time spent in contact with aerosols generated through mechanical agitation, aeration, screening and movement of wastewater during processing. We focused on fecal-oral transmission because various sources have reported that to be the most common way of wastewater worker exposure to biohazards (also called hand-to-mouth contact) [12,26–30]. This form of transmission is likely to occur as a result of improper hygiene during eating, drinking and smoking or by wiping the face with contaminated hands or gloves, or in extreme cases getting splashed in the face [27]. We identified five easy-to-use exposure categories with examples of routine tasks performed at WWTPs with Type A (no contact with wastewater anticipated) having the lowest risk and Type E (high risk) having the highest exposure level. Type E also included a subcategory (E1) due to anecdotal evidence provided by the focus groups suggesting that accidental splashes in the face do occur under rare occasions. Even though categories A and B should not involve direct contact with wastewater, focus group results highlighted that office workers may still be exposed to contaminated fomites through shared lunchrooms, washrooms, workspaces, objects and meeting rooms.

The only study that used our approach and defined risk categories was the work conducted by Zhang et al. (2021), which clustered workers into three groups: "low susceptibility", "high occupational susceptibility" and "high health susceptibility" [18]. Zhang et al. (2021) included health demographics of workers. Our objective was to provide practical mitigation strategies for workers without having to delve into their personal information as this is often protected at larger utilities. The results we provided also lend themselves well to integration into job safety assessments. A JSA, also called a job hazard analysis (JHA), is a procedure to integrate accepted safety and health principles and practices into a particular task or job operation. For each basic step of the job, potential hazards are identified, and the safest ways are recommended to do the task [25,30]. Given that smaller WWTPs have less resources and subject matter experts on H&S, the controls checklist we provide makes it easy for the average operator or plant manager to identify potential improvements to occupational health protection. The results also suggested that large WWTPs had different amounts of time allocated to tasks with varying levels of exposure to wastewater. Urban WWTP had the largest amount of FTE allocated to tasks in categories C > A > B > D > E, respectively. Industrial and municipal operations were more likely to allocate more FTE to tasks A > C > D > B > E, respectively. Luckily, high exposure risk tasks (Category E) required the least amount of time at all WWTPs, regardless of size.

Our data suggests that many of the QMRAs evaluating risk from wastewater exposures at large treatment plants may be overestimating risk by not evaluating risk by specific tasks [30] and not including WWTP size as a variable. Yan et al. (2021) reported that even when field engineers were equipped with KN90 masks, the health risks were still generally one order of magnitude higher than the WHO and U.S. EPA disability adjusted life years (DALY) benchmarks and that the infection risks for all WWTP staffs were generally two orders of magnitude over the benchmark [17]. Many others also reported unacceptable risk levels to wastewater workers [23,31,32]. The concerning work by Yan et al. (2021) and others contradicts work by Gui et al. (2022) and Kataki et al. (2022) who suggest that risk of infection can be reasonably mitigated through the use of engineering, administrative and PPE controls [16,20]. In addition, many systematic reviews have concluded that the epidemiological evidence supporting infections from wastewater work is insufficient to claim a causal association [33–35]. The present work will hopefully contribute to more QMRA refinement and less overestimation of risk.

The present work provides mitigation measures in addition to valuable information for water industry job safety assessments. However, it is important to note the limitations associated with this work. Exposure risks are site- and job-specific and time spent on specific tasks in this study may not be generalizable to all WWTPs. In addition, the type and level of hazards vary by wastewater type and by operating conditions, equipment, and configuration at each WWTP [30]. While these results provide a useful dataset that can

be used especially by smaller plants with limited resources, less skilled and experienced operators, no health and safety personnel and different safety challenges compared to larger systems [36], result applicability at new WWTPs will need to be validated further.

## 5. Conclusions

QMRA presents an opportunity to systematically assess risk to wastewater treatment plant (WWTP) workers and mitigate work-related infectious diseases using a structured and systematic approach. However, while QMRAs often explored the impacts of aeration or treatment mechanism (e.g., umbrella aeration tank versus microaeration, membranes) or the use of controls to mitigate risk such as ventilation and personal protective equipment (PPE), fewer studies addressed other variables, such as differing tasks across plants and time spent conducting these tasks or size of plant. QMRA approaches also vary substantially in their findings and recommendations. The objective of this paper is to provide a wastewater worker task characterization for urban (>300 megaliters per day (MLD)), municipal (10 MLD) and industrial (0.001 MLD) WWTPs along with appropriate measure for mitigating risk against fecal-oral or mucous membrane, contact transmission exposure. Routine tasks fell into five categories in ascending order of exposure and risk. Percentage full-time equivalents spent on each task category was estimated, along with amount of wastewater exposure (mL) and inhalation duration (h). Estimates were different between urban and municipal plants but were similar in industrial and municipal systems. Finally, the developed checklist helped identify potential mitigation measures and prioritize H&S solutions for all systems visited. The present work provides valuable and practical information for job safety assessments, H&S policies and QMRA method refinement.

**Funding:** This research received no external funding.

**Institutional Review Board Statement:** This study was conducted as part of a corporate health and safety assessment to mitigate risk of biological hazards exposures, in light of the West African Ebola outbreak and resulting heightened risk perception in the water industry in 2014-2015. The study did not require ethics approval. All participants were employees.

**Informed Consent Statement:** Not applicable.

**Data Availability Statement:** Not applicable.

**Conflicts of Interest:** The author declares no conflict of interest.

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
