# Peer review of "Protecting Wastewater Workers by Categorizing Risks of Pathogen Exposures by Splash and Fecal-Oral Transmission during Routine Tasks"

_waste, doi:10.3390/waste1010007_

Round 1

Reviewer 1 Report

Waste-2036967 Mitigating Risks of Pathogen Exposure at WTPs – reviewer comments

In this paper, the authors have identified inadequacies in previous analyses of risks to exposure of pathogens by WTP workers. Their aim is to use a quantitative approach (QMRA) to address this gap and to improve mitigation practices, especially in relation to Plant size. Their recommendations are presented in Table 4, as improvements to current industrial practice (Table 3).

In my view, the paper provides valuable new information to the industry and should be published. I offer the following comments to improve clarity and consistency of the narrative:

L53 – delete with

L60-62 – Sentence not clear – proves? – rephrase

L128-9 – In contrast to Table 2, The exposures are expressed as the amount of time spent in each activity in Full Time Equivalents (FTE)” and the next sentence refers to these total exposures for the Urban Plant. However, following that (L131-134), the discussion reverts to proportions (% - as shown in Table 2). I suggest the Authors should avoid switching the narrative between total numbers and proportions, as it can lead to confusion.

Fig 1. The order of bars L-R appears as Urban-Industrial-Municipal, whereas previous discussion and Table 2 were ordered logically by size Urban-Municipal-Industrial. The latter order would provide greater consistency.

Fig 1 refers to FTE as Full Time Employees, previously stated as Full Time Equivalents. Please clarify the definition of FTE on its first-time use.

L222-3 – The reason behind the statement “equipped with KN90 masks, the health risks were generally one order of magnitude higher”, is not clear. This point is of interest, since it seems counter-intuitive that using PPE (masks) increases risk - please clarify.

Reviewer 2 Report

In this nice manuscript, the author describes a method of categorizing risk by job function at wastewater treatment plants. This is an interesting discussion about exposure related to job and time doing specific tasks that may not be related to exposure. Specific comments:

- It is not clear that the title reflects what was done in this study. Perhaps consider another more descriptive title. 

- The abstract has a few places that need a grammar update. Lines 11 and 13 should be "explore" and "address" respectively.

- The font is not consistent in all of the tables, this can be fixed in an editorial review - but it would be good to address.

- The survey and literature review mentioned in section 3.4 and presented in Table 3 is not described in the materials and methods. As a result, it is unclear how Table 3 was compiled and how it relates to the rest of the study. 

- The focus groups used to form Table 4 are also unclear. This needs to be explained in the materials and methods. 

- A limitations section should be included in the discussion section. 

Round 2

Reviewer 2 Report

The authors have responded to reviewer comments.